# Effect of a Supervised Aerobic Exercise Training Program and Ginkgo Biloba Extract on Metabolic Parameters and Functional Capacity in HIV-Infected Subjects

**DOI:** 10.3390/healthcare13060663

**Published:** 2025-03-18

**Authors:** Raúl Soria-Rodríguez, Javier Méndez-Magaña, Nathaly Torres-Castillo, Erika Martínez-López, Edtna Jauregui-Ulloa, Juan López-Taylor, Cesar O. de Loera-Rodríguez, Ramón Sigala-Arellano, Fernando Amador-Lara

**Affiliations:** 1Instituto de Ciencias Aplicadas a la Actividad Física y Deporte, Departamento de Ciencias del Movimiento Humano, Educación, Deporte, Recreación y Danza, Centro Universitario de Ciencias de la Salud, Universidad de Guadalajara, Guadalajara 44340, Mexico; raul.soria@academicos.udg.mx (R.S.-R.); javier.mendez0173@alumnos.udg.mx (J.M.-M.); edtna.jauregui@academicos.udg.mx (E.J.-U.); taylor@cucs.udg.mx (J.L.-T.); 2Instituto de Nutrigenética y Nutrigenómica Traslacional, Departamento de Biología Molecular y Genómica, Centro Universitario de Ciencias de la Salud, Universidad de Guadalajara, Guadalajara 44340, Mexico; nathaly.torrescas@academicos.udg.mx (N.T.-C.); erika.martinez@academicos.udg.mx (E.M.-L.); 3Departamento de Fisiología, Centro Universitario de Ciencias de la Salud, Universidad de Guadalajara, Guadalajara 44340, Mexico; octavio.deloera@academicos.udg.mx; 4Laboratorio de Patología Clínica, Hospital Civil de Guadalajara “Fray Antonio Alcalde”, Guadalajara 44280, Mexico; rsigala@hcg.gob.mx; 5Departamento de Clínicas Médicas, Centro Universitario de Ciencias de la Salud, Universidad de Guadalajara, Guadalajara 44280, Mexico

**Keywords:** physical activity, exercise training, HIV infection, Ginkgo biloba, statins

## Abstract

**Background**: A remarkable increase in metabolic comorbidities occur in people living with HIV infection (PLWH). Supervised physical activity provides significant health benefits. Ginkgo biloba (GKB) extract has been reported to have a wide range of metabolic advantages. This study aimed to examine the effects of an exercise training (ET) program and a GKB extract on PLWH. **Methods**: This was a randomized placebo-controlled double-blind study. Twenty-eight PLWH were assigned to receive a placebo (*n* = 10), GKB extract (*n* = 10), or statins (*n* = 8). All patients underwent a supervised ET program 3–5 times per week. Anthropometric measurements, functional capacities, and metabolic parameters were assessed in all participants at baseline and after 12 weeks of follow-up. **Results**: After the 12-week intervention, body fat decreased significantly by 2–3% in all groups relative to their baseline values (*p* < 0.05). Total cholesterol and LDL-c were significantly decreased in the ET + statin group (*p* = 0.04, and *p* = 0.007, respectively) compared to baseline values, while HbA1c and the HOMA-IR index were significantly decreased in the ET + GKB group (*p* = 0.03 and *p* = 0.02, respectively) compared to baseline values, and a significant increase in CD4+ T cell mean was observed in the ET + placebo group (*p* = 0.005) compared to baseline values. A significant increase in cardiorespiratory capacity (VO_2max_) from their baseline values was observed in all groups (*p* < 0.001) after 12 weeks of intervention from their baseline values. **Conclusions**: Body fat and cardiorespiratory fitness significantly improved after a 12-week supervised ET program. GKB extract significantly decreased insulin resistance.

## 1. Introduction

An increase in the life expectancy of people living with HIV (PLWH) due to antiretroviral therapy (ART) has led to an increase in the burden of non-communicable diseases in this population [1]. PLWH have a high prevalence of obesity-related comorbidities, including dyslipidemia, type 2 diabetes mellitus (T2DM), hypertension, and chronic kidney disease [2]. The risk of cardiovascular disease (CVD) is 2-fold higher in PLWH [3], and some authors consider that HIV infection should be considered as a major risk factor along with other, traditional risk factors [4]. Immune activation and chronic low-grade inflammation caused by HIV infection, in addition to the adverse effects of ART, contribute to the increased risk of CVD [5]. In a study of PLWH at low to moderate risk for CVD, without known CVD, aged 40 to 75, and receiving stable ARVs, 49% were found to have coronary plaque; markers of inflammation (IL-6 and LpPLA2) were associated with coronary artery disease independent of traditional risk factors [6]. Currently, non-AIDS-related mortality rates are largely attributable to CVD [7].

Weight gain with contemporary ART based on integrase strand transfer inhibitors (INSTIs)-containing regimens and tenofovir alafenamide (TAF) has been well documented [8,9,10]. The proportion of metabolically unhealthy individuals is higher in people with HIV with excessive weight and central obesity [11]. INSTI use has been associated with increased risk of new-onset diabetes mellitus/hyperglycemia in the 6 months following ART initiation [12].

Studies have shown that physical activity (PA) has multiple health benefits for PLWH, including cardiovascular health [13], functional capacity [14], improvements in metabolic parameters [15], body muscle strength [16], body composition [17], and quality of life (QoL) [18]. Aerobic exercise has been shown to increase VO_2_ peaks [17,19], while resistance exercise and combined (aerobic and resistance) exercise have been shown to decrease markers of inflammation [20], improve lipid profiles [21], and increase in-bone mineral density [22], with contradictory results on changes in CD4+ T cell count in PLWH [17,21]. However, it has been reported that PA is lower in individuals with HIV than in most other populations with chronic diseases, and more than half of PLWH do not meet the physical activity recommendations of the World Health Organization (WHO) of at least 150 min of moderate-intensity physical activity per week [23].

Cardiometabolic disease management strategies, including the use of medicinal herbs, have been used worldwide. *Ginkgo biloba* (GKB), an herb used in traditional Chinese medicine for thousands of years, has high medicinal value because it contains flavonoids, treptene lactones, and phenolic compounds [24]. GKB has various properties, including antioxidants, free radical scavenging, membrane-stabilizing, anti-inflammatory, anti-platelet activating factors, antihypertensive, vasodilatory, cardioprotective, neuroprotective, anti-apoptotic, and anticancer activities [24] and has been used to treat atherosclerosis, ischemic heart disease, dementia, cerebrovascular insufficiency, hypertension, and peripheral arterial occlusive disease [25,26,27].

Experimental and clinical studies have reported metabolic benefits in hypertriglyceridemia and hypoglycemic properties; therefore, GKB supplementation has been used to treat dyslipidemia and to prevent or treat T2DM [28,29,30]. However, to our knowledge, GKB has never been used in the management of metabolic disorders in PLWH.

Considering the increased prevalence of metabolic abnormalities in PLWH, the aim of the study was to investigate the effect of a supervised exercise training program alone or in combination with supplementation of GKB or statins on functional capacity, body composition, and metabolic parameters after 12 weeks of follow-up. The study hypothesizes that exercise training (ET) in combination with GKB supplementation improves functional capacity and metabolic parameters in PLWH.

## 2. Materials and Methods

### 2.1. Design and Approval of the Study

This was a single-center, randomized, double-blind, placebo-controlled, parallel group study, with a three-arm group, conducted in Guadalajara, Jalisco, Mexico, in accordance with the ethical principles of the World Medical Association for Medical Research involving Human Subjects. This study was approved by the Ethics Committee of the Hospital Civil de Guadalajara (no. 165/21) and was registered at ClinicalTrials.gov (identifier NCT06403787). The purpose of the study was explained to the participants, and written informed consent was obtained before screening and data collection.

The eligibility criteria for all groups were as follows: HIV-infected men aged 18–55 years on ART with coformulated bictegravir, emtricitabine, and tenofovir alafenamide (B/F/TAF), with at least six months of undetectable HIV viral load (<50 copies/mL), CD4^+^ T cell count >200 cells/µL, and mixed dyslipidemia with two or more abnormal findings: total cholesterol ≥ 200 mg/dL, LDL-c ≥ 100 mg/dL, HDL-c ≤ 40 mg/dL, and triglycerides ≥ 150 mg/dL. The exclusion criteria were subjects with any illness or physical disability that prevents them from carrying out physical training, active hepatitis B and C infection, heavy alcohol use (≥15 drinks per week), known hypersensitivity to Ginkgo biloba extract or statins, and the use of any dietary supplement within 30 days of study enrollment.

### 2.2. Randomization and Blinding

After eligibility was confirmed, the patients were randomized into three groups (allocation ratio of 1:1:1) using computer-generated random numbers to allocate GKB extract 120 mg QD, atorvastatin 20 mg QD, or placebo (Figure 1).

### 2.3. Procedures

All subjects participated in a supervised aerobic ET program 3–5 times per week for 12 weeks. Exercise intensity gradually increased; in the first 4 weeks, the intensity was 50–60% heart rate maximum (HRmax), and in the next 8 weeks, it increased to 60–80%. The duration of the exercise was 50 min per session; each exercise session included a 5 min warm-up and 5 min cool-down period with stretching exercises. The sessions were conducted in a gym close to the center and were supervised by an exercise researcher.

### 2.4. Funcional Capacity Measurements

At baseline and after 12 weeks, functional capacity was assessed using the following tests:

**Cardiorespiratory fitness.** The McArdle step test was used to determine VO_2Max_ during an endurance test. This test consists of going up and down a 41 cm step for 3 min; each up and down cycle consists of 4 repetitions. Women perform 22 cycles/min (metronome at 88 wpm—4/4) and men perform 24 cycles/min (metronome at 96 wpm—4/4). At the end of the 3 min, the subject remains standing, and the HR is recorded for 15 s (from the 5th to the 20th second of recovery). The value obtained is multiplied by 4 to obtain the HR value during the first minute of recovery (HR1). Our equations are applied to obtain the VO_2Max_ (ml/min × Kg).

**Strength.** Back-leg and hand-held strength were evaluated using a dynamometer (Takei III Smedley type digital dynamometer^®^ manufactured in Niigata Japan by Takei Scientific Instruments Co., Ltd.). The following evaluations were carried out: *Strength in hand*—This test was performed to measure the muscular power of the flexor muscles of the arm and forearm. The person evaluated stands up and holds the dynamometer with his hand, holding it as firmly as possible with fingers. The arm is slightly bent and remains at the side of the body; the palm of the hand should be placed towards the thigh without touching it. At the signal of the evaluator, the subject applies maximum pressure to the dynamometer with the greatest possible force. During the execution, the participant is not allowed to shake the equipment, modify the posture of their body, or the position of the dynamometer. The highest reading of 2 attempts is recognized. *Strength in back*—This test aims to measure the muscular power (static strength) of the back muscles. To perform it, the subject stands on an extension dynamometer. The person evaluated remains with his legs straight and shoulder-width apart, and leans his trunk slightly forward, while holding the handle, connected to the dynamometer through a chain, with both hands. The subject maintains, always, extension of their arms and legs, so that the effort can only be made with extension of the trunk muscles. At the signal of the evaluator, the back must be extended, stretching the dynamometer with as much force as possible. *Strength in legs*—The purpose of this test is to measure the strength of the leg muscles. The subject stands on the extension dynamometer. The performer remains with the legs semi-flexed and separated at shoulder level, with the trunk completely straight, while holding, with both hands, a straight bar connected to the dynamometer through a chain. The person tested must keep their back and arms extended throughout the entire test, so that only an effort can be made with the leg muscles. At the signal of the evaluator, the performer performs a powerful leg extension to stretch the dynamometer with the maximum possible power.

**Flexibility.** A sit and reach test was used. The test is performed by placing the subject facing the widest side of a box. The participant’s feet must be fully supported on the box, and the trunk is bent forward without bending the legs. The arms are extended to the maximum to move the ruler as far as possible. The highest measurement that the subject can achieve in more than 2 s is taken. No stretching is allowed, and only 2 attempts are allowed.

**Balance.** We performed the single leg static balance test; this test helps determine static balance on one foot. The evaluated person stands with his feet shoulder-width apart with his back straight, and he is asked to close his eyes and raise the foot of his choice backwards at a right angle. The evaluator counts the number of times that the subject attempts to lower his foot trying to maintain this position in one minute).

In addition, anthropometric measurements were evaluated in all participants. Each subject was given a diary to record medication adherence and report adverse events. All subjects received dietary counseling. Each participant attended visits every 4 weeks to the center to evaluate adherence to the ET program, intake of the study medication, reporting of adverse events, changes in functional capacity through the previously mentioned physical tests and anthropometric measurements, and completion of the IPAQ short-version questionnaire.

### 2.5. Blood Sample Collection and Processing

Laboratory tests to assess metabolic profiles were performed in a fasting state at the baseline visit and week 12. Fifteen mL of peripheral whole blood was collected and distributed into three pre-treated tubes as follows: 1. A 6 mL EDTA-coated tube was used for complete blood count and CD4+ T-cell quantification via flow cytometry (Beckman Coulter AQUIOS^®^, manufactured at Beckman Coulter’s facility in Miami, FL, USA). Plasma aliquots were used to determine the viral load via nucleic acid testing (NAT RT-PCR) (Abbott ALINITY m HIV-1 ASSAY^®^, manufactured by Abbott Laboratories in Abbott Park, IL, USA). 2. A 6 mL plain serum tube (without anticoagulant) was used for biochemical analyses, including hematology, insulin, glucose, HbA1c, creatinine, urea, total cholesterol, LDL-c, HDL-c, VLDL-c, triglycerides, total bilirubin, AST, ALT, GGT, and alkaline phosphatase, assessed using photometric and potentiometric methods (Abbott ALINITY c ^®^, produced by Abbott Laboratories manufactured in Lake County, IL, USA). 3. A 3 mL sodium citrate-coated tube was used for coagulation studies (prothrombin time, activated partial thromboplastin time, fibrinogen, INR) via coagulometric, chromogenic, and immunological assays (Instrumentation Laboratory ACL Top 750 CTS, manufactures by Werfen in Bedford, MA, USA).

### 2.6. Statistical Analysis

Sample size was calculated based on the results of Smith BA et al. [31], who found a significant difference in the BMI after 12 weeks of the supervised exercise program. We use the G*Power software version 3.1.9.7 to calculate a priori required sample size, considering an α = 0.05, a power of 0.80, and an effect size of 1.63; therefore, an *n* = 8 subjects per group was obtained. The Shapiro–Wilk test was used to analyze the normality of the distribution of the quantitative variables. Non-normally distributed data were analyzed using non-parametric statistical tests; contrary, parametric tests were used for normally distributed variables. Quantitative variables are expressed as mean ± standard deviation (SD) or median (interquartile range); qualitative variables are expressed as frequency (*n*) and percentage (%). One-way ANOVA or Kruskal–Wallis tests were used to compare quantitative variables between the three study groups. Student’s *t* test for repeated samples or the Wilcoxon test was used to compare final and initial intragroup values. Data were analyzed using SPSS version 20. Statistical significance was set at *p* < 0.05.

## 3. Results

Twenty-eight patients with HIV were included in this study. All the patients underwent supervised ET. Ten patients received placebo (ET + Placebo), ten received GKB extract (ET + GKB), and eight patients received statin (ET + Statin). B/F/TAF was the first regimen in 42% of the subjects, with a mean duration of 3 years in this regimen; 58% of subjects had already received previous regimens. When comparing the baseline demographic variables between the study groups, significantly higher levels of ALT, urea, and BUN were found in the ET + Statin group than in the ET + GKB group (*p* < 0.05). No other significant differences were observed between groups (Table 1).

No significant differences were observed in the anthropometric variables after a 12-week intervention between the three groups. The effect on metabolic variables after the 12-week intervention showed a significant decrease in total cholesterol in the group receiving statins compared to that in the other two groups (*p* < 0.05) (Table 2).

Body fat decreased significantly by 2–3% at the end of the 12-week intervention in all three groups relative to their baseline values (*p* < 0.05) (Figure 2A). Waist circumference (WC) also decreased significantly in participants receiving statins compared with baseline values (*p* = 0.009) (Figure 2B).

Total cholesterol and LDL-c were significantly decreased in the ET + statin group at the end of the study (*p* = 0.049 and *p* = 0.007, respectively) compared to baseline values (Figure 2C,D), while HbA1c (Figure 2E) and the HOMA index (Figure 2F) were significantly decreased in the ET + GKB group (*p* = 0.03 and *p* = 0.02, respectively) compared to baseline values, and a significant increase in CD4+ T cell mean was observed in the ET + Placebo group (*p* = 0.005) (Figure 2G) compared to baseline values.

A significant increase in cardiorespiratory capacity (VO_2 max_) from their baseline values was observed in all three groups (*p* < 0.001) after 12 weeks of intervention from their baseline values, in addition to a significant increase in back strength in the ET + GKB group (*p* = 0.02) and flexibility in the ET + statin group (*p* = 0.02) (Table 3).

## 4. Discussion

We examined the effect of a supervised exercise training (ET) program, alone or in combination with supplementation of GKB or statin use, on body composition, functional capacity, and metabolic parameters after 12 weeks of follow-up. Overall, we found a significant decrease of 2–3% body fat and a significant increase in cardiorespiratory fitness (VO2max) observed in all groups, mirroring the results of ET. Additionally, an increase in CD4+ T cells was observed in the group that only performed ET. A significant improvement in total cholesterol was found in the ET + statin group compared to the other two groups, while a significant improvement in HbA1c and insulin resistance (HOMA-IR) was found in the ET + GKB group compared to their baseline values. To our knowledge, this is the first evidence of the effects of GKB on improving glucose control and insulin resistance.

The prevalence of chronic comorbidities in PLWH is high [32], and a strategy that has demonstrated health benefits in this population is PA [17]. There is evidence that PA reduces the risk of chronic diseases including obesity, T2DM, osteoporosis, breast and colon cancer, and coronary artery disease in the general population [33]. Studies on supervised and unsupervised PA have shown greater benefits when the activity is supervised [34], and supervised PA has been reported to improve functional capacity [14] and adherence in PLWH [35]. We found a significant reduction in total body fat with supervised aerobic ET in all groups, which is in accordance with other studies that have reported improvements in body fat and body composition in PLWH with supervised aerobic PA [31,36]. Resistance exercise and combined aerobic and resistance exercise have also demonstrated benefits in terms of body fat, body composition, muscle strength, cardiorespiratory fitness, and quality of life in PLWH [14,20,37,38].

PA has also shown multiple metabolic benefits, including improvements in insulin resistance, blood lipid levels, and hepatic fat content, regardless of weight loss [39]. A reduction in the risk of diabetes with significant improvements in glucose and insulin, both fasting and postprandial, HOMA-IR, weight, systolic blood pressure, and triglycerides were found in a 6-month intervention study with dieting and physical activity in PLWH and impaired fasting glucose [40]. Similarly, advanced glycation end products (AGEs) (proteins or lipids that become glycated because of exposure to reduced sugars) are implicated in the risk of the development of cardiovascular disease, diabetes, and other chronic diseases; these AGEs that promote the activation of inflammatory and procoagulant pathways that may contribute to CVD were significantly decreased after a supervised training program of combined exercise (aerobic, resistance, and flexibility) for three months in physically inactive HIV-infected subjects [15]. In our study, we did not find a decrease in HOMA-IR or HbA1c in the ET + P group compared to the ET + S group; however, in the ET + GKB group, there was a significant improvement in both parameters.

A significant increase in VO_2_ peak was found by Mutimura et al. in a 6-month study of supervised PA in PLWH [36], which is in line with our findings of a significant increase in VO_2_ max observed in the three groups, indicating an improvement in cardiorespiratory fitness through the 12-week ET program. However, one study in older HIV-infected men found that only high-intensity aerobic exercise and not moderate-intensity aerobic exercise for 16 weeks was associated with a significant improvement in cardiorespiratory fitness (VO_2_ peak) [19].

Several studies in HIV− and HIV+ subjects have demonstrated that exercise has anti-inflammatory effects by reducing inflammatory biomarkers, including IL-1β, IL-6-IL-8, TNFα, and improving immune function [16,20,41]. Nevertheless, a study of PA in PLWH found no differences in the markers of immune activation (CD38 and HLA-DR) or inflammation (IL-6 and TNF-α), although exercise was self-prescribed [42]. There are contradictory results regarding the improvements in immune function associated with physical activity in PLWH. Smith et al. found no significant changes in CD4^+^ T cell counts in an RCT after 12 weeks of aerobic exercise in PLWH [31], whereas Brito-Neto et al. found that a 12-week resistance training program resulted in a significant increase (15.7%) in CD4^+^ T cell counts [37]. A significant increase in the number of CD4^+^ T cells was observed in the ET+ P group.

The prevalence of T2DM among PLWH is up to four times higher than that among the general population in some regions of the world [43]. Several risk factors have been identified for the development of T2DM in older PLWH, including the duration of HIV infection, lower CD4^+^ T cell nadir, long durations of HIV infection, use of older-generation antiretroviral therapy, high BMI, and arterial hypertension [44].

GKB exerts antidiabetic effects by increasing insulin expression and sensitivity. GKB extract increases pancreatic β-cell function [45] and improves insulin sensitivity by enhancing IRS-2 transcription [46], as IRS-2 is a crucial element in insulin signaling, and studies have found that a deficiency of IRS-2 causes insulin resistance [47]. We found a significant decrease in insulin resistance (HOMA-IR) and HbA1c levels in our study of non-diabetic PLWH in the ET + GKB group; therefore, its usefulness as a dietary supplement for the prevention of diabetes should be considered. Aziz et al. found a significant decrease in HbA1c, fasting serum glucose, BMI, WC, and VAI in T2DM subjects ineffectively treated with metformin, to whom GKB extract supplementation was added for six months [30]. The effects on BMI, WC, and VAI have been associated with an increase in lipolysis induced by the GKB extract [48].

A pilot study of subjects with metabolic syndrome (MS) found a significant decrease in hs-CRP and HOMA-IR, as well as in other inflammation and oxidative stress biomarkers and nanoplaque formation, with the administration of GKB extract over 2 months. The HOMA-IR score and nanoplaque formation were significantly correlated in this study [49]. HOMA-IR predicts incidental symptomatic CVD independent of classic risk factors and several blood biomarkers; therefore, insulin resistance should be an important target not only for reducing or treating T2DM, but also for reducing cardiovascular risk [50]. HOMA-IR has been correlated with increased CVD/total mortality in both the diabetic and non-diabetic populations [51,52]. Furthermore, in the MESA study, HOMA-IR was found to predict the incidence and progression of coronary artery calcification, although not independently of MetS status [53].

Experimental studies with GKB extract have observed other metabolic benefits, including reductions in body adiposity in diet-induced-obesity rats, restoration of obesity-induced insulin signaling impairment [54], inhibition of adipogenesis, regulation of lipid metabolism, body weight reduction in mice [55], and a decrease in adipocyte volume from obese rats to dimensions equivalent to adipocytes from non-obese rats, suggesting a potential anti-obesogenic effect of GKB [56]. Moreover, the GKB extract exerts several lipid-lowering effects, including decreased cholesterol absorption, inactivation of HMG-CoA, and improvement of essential polyunsaturated fatty acids [57]. The favorable effects of GKB on lipids have been observed in both experimental and clinical studies [58,59]. We did not find any relevant changes in the lipid parameters of our patients in the GKB group, probably because of the short study duration. However, the statin-treated group showed significant changes in total cholesterol and low-density lipoprotein cholesterol levels. These results agree with those of a study carried out by Zanetti et al., who found that ET, statin use, and the combination of both decreased total cholesterol, LDL-c, TG, CRP, IL-1-β, and carotid intima-media thickness compared with no intervention in PLWH [60]. Our study has limitations, including the small sample size; it was carried out in a single center and a single country, and only males were enrolled; therefore, the results cannot be generalized to other regions or to females. Analysis of inflammatory biomarkers would have strengthened the results of this study.

## 5. Conclusions

HIV infection is now a treatable chronic disease, and it is a concern that about half of PLWH are physically inactive. Our study demonstrated that supervised physical activity improves body fat and cardiorespiratory capacity. This effective non-pharmacological intervention should be encouraged in PLWH to engage them in physical activity. In contrast, GKB extract is an inexpensive supplement with effects on insulin resistance in PLWH and could be used in the prevention and add-on treatment of T2DM. Further large-scale studies are required to confirm these findings.

## Figures and Tables

**Figure 1 healthcare-13-00663-f001:**
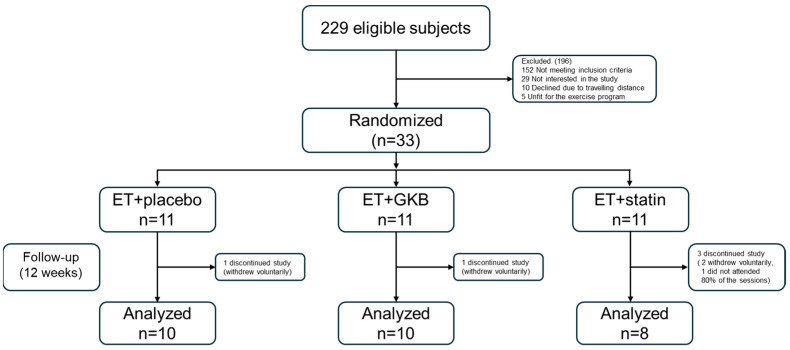
Flow chart of the participants in the study. ET, exercise training; GKB, Ginkgo biloba.

**Figure 2 healthcare-13-00663-f002:**
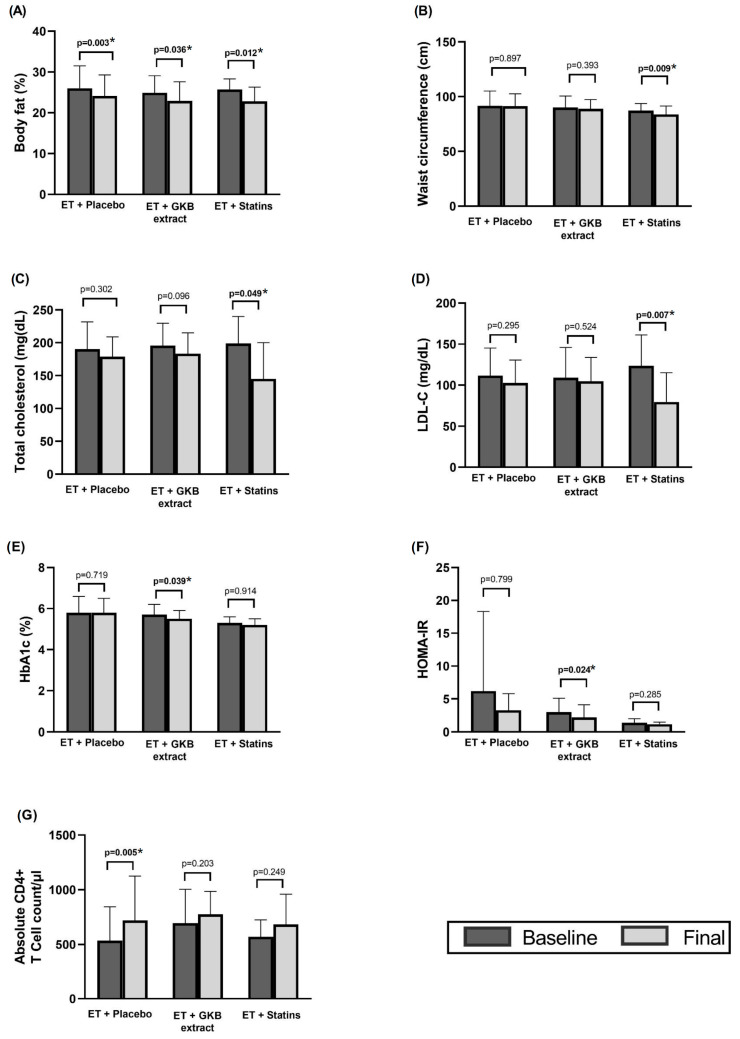
Intragroup changes in body composition (**A**), waist circumference (**B**), total cholesterol (**C**), LDL-C (**D**), HbA1c (**E**), HOMA-IR (**F**), and absolute CD4+ T cell count /µL (**G**) after 12 weeks of intervention. * *p* < 0.05.

**Table 1 healthcare-13-00663-t001:** Basal characteristics of participants.

Characteristics	ET + Placebo *n* = 10	ET + GKB *n* = 10	ET + Statin *n* = 8	*p* Value
Mean age, years (SD)	41.8 ± 11.0	41.8 ± 7.0	40.5 ± 10.5	0.949
Weight (kg)	80.7 ± 13.3	75.9 ± 12.4	76.2 ± 12.5	0.277
Waist circumference (cm)	91.6 ± 13.6	90.0 ± 10.5	87.2 ± 6.5	0.695
Waist-to-hip ratio	0.90 ± 0.10	0.95 ± 0.07	0.90 ± 0.05	0.253
Waist-to-height ratio	0.53 ± 0.08	0.53 ± 0.08	0.52 ± 0.04	0.906
BMI (kg/m^2^)	28.0 ± 4.1	26.6 ± 4.2	25.7 ± 2.6	0.854
Sum of 8 skinfolds (mm)	118.8 ± 47.7	127.6 ± 40.8	129.6 ± 43.9	0.855
Body fat mass (%)	26.0 ± 5.5	24.9 ± 4.2	25.7 ± 2.6	0.854
Lean body mass (%)	39.0 ± 6.4	40.3 ± 4.9	40.2 ± 3.4	0.827
Bone mass (%)	13.7 ± 1.4	14.3 ± 1.5	14.2 ± 0.8	0.560
Absolute CD4+ T Cell count/μL, mean	534.3 ± 310.4	694.2 ± 310.1	570.6 ± 154.6	0.171
HIV-1 RNA (copies/mL), mean	34.5 (28.7–39.0)	31.0 (31.0–34.2)	35.0 (31.0–39.0)	0.394
Total billirrubin (mg/dL)	0.80 ± 0.4	0.63 ± 0.2	0.58 ± 0.2	0.683
Direct billirrubin (mg/dL)	0.14 ± 0.05	0.11 ± 0.03	0.11 ± 0.06	0.335
Total protein (g/dL)	7.2 ± 0.5	7.2 ± 0.4	7.1 ± 0.3	0.933
Albumin (g/dL)	4.2 ± 0.1	4.2 ± 0.2	4.3 ± 0.2	0.518
Globulin (g/dL)	3.1 ± 0.5	2.9 ± 0.4	2.7 ± 0.4	0.374
Alanine aminotransferase (IU/L)	31.8 ± 11.2	36.8 ± 55.0	18.7 ± 5.4	0.024 *
Aspartate aminotransferase (IU/L)	22.0 ± 5.9	30.1 ± 29.0	17.2 ± 4.5	0.061
Gamma-glutamyl transpeptidase (IU/L)	45.8 ± 31.0	35.7 ± 32.4	27.3 ± 15.7	0.273
Alkaline phosphatase (IU/L)	79.7 ± 21.4	73.6 ± 14.8	69.2 ± 19.5	0.533
Lactate dehydrogenase (U/L)	159.1 ± 35.7	174.1 ± 41.2	156.8 ± 22.7	0.514
Urea (mg/dL)	34.0 ± 10.9	28.3 ± 7.7	41.2 ± 11.2	0.039 *
BUN (mg/dL)	15.8 ± 5.0	13.2 ± 3.5	19.3 ± 3.5	0.035 *
Creatinine (mg/dL)	0.91 ± 0.12	0.096 ± 0.12	1.04 ± 0.19	0.213
Prothrombin time	10.8 ± 0.9	11.0 ± 0.4	10.7 ± 0.6	0.824
INR	0.97 ± 0.09	1.00 ± 0.04	0.93 ± 0.08	0.258
Partial thromboplastin time	32.4 ± 2.5	33.6 ± 4.5	31.4 ± 2.7	0.487
Fibrinogen (mg/dL)	421.4 ± 73.9	459.0 ± 93.6	422.6 ± 82.9	0.559

Abbreviations: ET, exercise training; GKB, Ginkgo biloba; SD, standard deviation; BMI, body mass index; INR, international normalized ratio. Data are presented as mean ± SD or median (interquartile range). A one-tailed analysis of variance (ANOVA) was used to compare groups and Bonferroni for post hoc comparisons. Variables non-normally distributed were analyzed with the Kruskal–Wallis, besides for informational purposes, their values are reported as median (IQR) and as mean ± SD. * *p* < 0.05, comparing the ET + statin group vs. the ET + GKB group.

**Table 2 healthcare-13-00663-t002:** Metabolic changes between study groups after 12-week intervention.

Characteristics	ET + Placebo *n* = 10	ET+ GKB *n* = 10	ET+ Statin *n* = 8	*p* Value
Total cholesterol (mg/dL)	−11.5 ± 33.2	−12.3 ± 20.9	−53.7 ± 63.9	0.044 *
HDL cholesterol (mg/dL)	0.5 ± 9.1	2.1 ± 3.5	1.8 ± 7.9	0.882
LDL cholesterol (mg/dL)	−8.9 ± 25.3	−4.3 ± 20.5	−44.0 ± 32.5	0.426
Triglycerides (mg/dL)	−9.5 ± 93.9	−27.0 ± 90.1	−0.8 ± 141.5	0.522
Glucose (mg/dL)	−5.9 ± 22.2	−2.4 ± 11.1	−2.6 ± 19.6	0.687
Insulin (µU/mL)	−4.4 ± 20.5	−3.0 ± 2.7	−0.9 ± 2.2	0.107
Glycated hemoglobin A1c (%)	−0.0 ± 0.2	−0.1 ± 0.2	0.0 ± 0.2	0.340
HOMA-IR ^†^	−2.9 ± 10.0	−0.7 ± 0.7	−0.2 ± 0.5	0.119
Triglycerides/HDL-c ratio	0.1 ± 2.6	−1.4 ± 3.0	−0.6 ± 3.4	0.903

Abbreviations: ET, exercise training; GKB, Ginkgo biloba; HOMA-IR, Homeostatic Model Assessment of Insulin Resistance. Data are presented as mean ± SD or median (interquartile range). A one-tailed analysis of variance (ANOVA) was used to compare groups and Bonferroni for post hoc comparisons. Variables with † symbol were non-normally distributed and were analyzed with the Kruskal–Wallis test; for informational purposes, their values are reported as median (IQR) and as mean ± SD. * *p* < 0.05 comparing ET + statin group vs. ET + GKB group.

**Table 3 healthcare-13-00663-t003:** Intragroup changes in functional capacity pre- and post-12-week intervention.

Variable	ET + Placebo *n* = 10			ET + GKB *n* = 10			ET + Statin *n* = 8		
	Basal	Final	*p* Value	Basal	Final	*p* Value	Basal	Final	*p* Value
VO2Máx ^†^	47.9 ± 5.3	59.9 ± 8.9	0.001 **	46.5 ± 10.1	55.0 ± 7.4	0.002 **	45.0 ± 4.2	58.9 ± 7.0	0.000 ***
Grip strength in non-dominant hand (Kg)	31.2 ± 5.4	33.3 ± 9.0	0.333 ^†^	61.9 ± 90.1	33.4 ± 4.9	0.674 ^†^	34.2 ± 8.6	346 ± 6.4	0.838
Grip strength in dominant hand (Kg)	33.7 ± 8.6	33.3 ± 11.9	0.849	34.4 ± 4.1	35.6 ± 6.1	0.380	35.6 ± 6.1	37.8 ± 6.1	0.297
Back strength (Kg)	36.5 ± 20.7	42.8 ± 18.1	0.272	32.6 ± 11.4	40.0 ± 12.9	0.025 *	39.3 ± 18.1	39.2 ± 15.0	0.978
Lower limb strength (Kg)	37.7 ± 13.6	42.8 ± 16.0	0.434	35.4 ± 14.3	40.0 ± 14.7	0.161	36.5 ± 18.2	39.3 ± 14.3	0.635
Flexibility (cm)	24.3 ± 10.4	24.5 ± 10.1	0.908	16.1 ± 7.4	17.5 ± 8.3	0.608	19.3 ± 7.8	25.0 ± 7.5	0.027 *
Balance	6.2 ± 5.1	3.3 ± 3.1	0.053	2.3 ± 3.3	1.5 ± 2.5	0.461 ^†^	5.6 ± 5.7	4.7 ± 5.5	0.570

Abbreviations: VO_2_ max, maximum volume of oxygen, SD, standard deviation; Data are presented as mean ± SD or median (interquartile range). Intragroup comparisons were made with Student’s T test for pared samples or the Wilcoxon test according to the normality of the variables (identified with the symbol †). For informational purposes, the values of non-normally distributed variables are reported as median (IQR) and as mean ± SD. * *p* < 0.05, ** *p* ≤ 0.005, *** *p* ≤ 0.001 comparing basal vs. final in each group.

## Data Availability

All relevant data are within the paper.

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
