# Peer review of "Effect of a Supervised Aerobic Exercise Training Program and Ginkgo Biloba Extract on Metabolic Parameters and Functional Capacity in HIV-Infected Subjects"

_healthcare, 2025, doi:10.3390/healthcare13060663_

Round 1
Reviewer 1 Report
Comments and Suggestions for Authors
Abstarct
Line 40: ‘’Future studies with larger sample sizes are required.’’ This sentence should be presented as a suggestion at the end of the discussion. Instead, a recommendation or conclusion appropriate to the research findings should be given in abstract.
Introduction
The introduction should be improved.
The molecular mechanisms between the previously examined parameters and HIV should be addressed more comprehensively. From now on, the unique value and importance of the research should be revealed and clearly expressed. Then, the research question or hypothesis should be stated. The last purpose sentence should be written.
Materials and Methods
The method has many shortcomings:
How was the number of participants in the study decided? Power analysis should be done.
All characteristics and details of the participants should be given.
Which step test was used? Give details.
Which brand of equipment was used for the force tests? What was the procedure like?
Which balance test was performed and how?
How much blood was taken from the participants? What type of tubes was it taken into?
Which brand of devices were analyzed using?
If non-parametric tests were used, why were evaluations made based on means and standard deviations?
Results
Does the p value given in Table 2 belong to ANOVA or Wilcoxon test? It is not understood and should be reconsidered.
Why were only the parameters in figure 2 explained with figures? Also, why are there only within-group comparisons in this figure? It should be rethought.
Table 3's writing style is different from others.
Discussion
The purpose can be reminded when starting the discussion. The main findings of the study are summarized. Then it is written whether the research hypothesis is confirmed or not. Then the discussion begins.
Comments on the Quality of English LanguageIt is normal.
Reviewer 2 Report
Comments and Suggestions for Authors
The study is overall interesting, despite the short time of observation and the small population examined. Materials are adequately exposed and the references are pertinent. As it is known that PLWH have an increased risk for cardiovascular events and inflammation markers, it is important to explore solutions to improve quality of life and long term risk in these patients.
I have, though, some concerns and suggestions:
PLWH are proven to have an increased risk for cardiovascular events, which I believe is a significant background for your research. I suggest adding some lines in the introduction about the increased risk for CVD in PLWH.
The study examined a population of PLWH on ART with B/F/TAF only. Was it a choice to specifically analyze only people in therapy with this regimen? If so, I believe it would be better to not generalize, as both the conclusion and title refer to PLWH in general, but it should be mentioned that the regimen analyzed was only B/F/TAF.
Please clarify if levels of HIV-1 RNA reported in Table 1 refer to the ones at the diagnosis or at the first visit. Because an undetectable viraemia for at least six months is among the inclusion criteria (please, define undetectable and which cut-off you're considering), but the mean levels reported especially in ET + GKB group do not classify as undetectable. And if they refer to the first visit, I wanted to ask if HIV viral load was only considered at baseline or was it analyzed at the 12w visit.
Change in CD4+ T cell should be reported in the Table too, as it is mentioned that there was a statically significant improvement after 12 weeks.
SCOLTA study reports an increase in risk of cardiovascular events in PLWH on ART with Integrase Strand-Transfer Inhibitors especially in the first two years after the exposure to INSTIs. Therefore it would be interesting to know if PLWH in your study were on first line treatment with B/F/TAF and how long or had they experienced previous regimen.
I suggest revising lines 197 to 201 and maybe separate the two concepts (one about glycation products and one about triglycerides).
Lines 264 to 267 must be revised as it sounds redundant with previous lines and it is unclear whether this data refers to your study or not.
Round 2
Reviewer 1 Report
Comments and Suggestions for Authors Thank you for your revisions. Your research hypothesis is still missing in the introduction. I suggest you add it. Good work.Author Response
Please see the attachment.

Reviewer 2 Report
Comments and Suggestions for Authors
All my previous concerns have been adequately addressed and the quality of the paper significantly improved.
